# The Surveillance of *Borrelia* Species in *Camelus dromedarius* and Associated Ticks: The First Detection of *Borrelia miyamotoi* in Egypt

**DOI:** 10.3390/vetsci10020141

**Published:** 2023-02-10

**Authors:** Radwa Ashour, Dalia Hamza, Mona Kadry, Maha A. Sabry

**Affiliations:** Department of Zoonoses, Faculty of Veterinary Medicine, Cairo University, Giza 12211, Egypt

**Keywords:** 16S-23S rRNA, glpQ, *Borrelia burgdorferi*, *Borrelia miyamotoi*, camels

## Abstract

**Simple Summary:**

Lyme borreliosis (LB) is a zoonotic disease caused by the spirochete *Borrelia burgdorferi* sensu lato (s.l.) transmitted to humans by the bite of infected Ixodes ticks. *Borrelia miyamotoi* is a spirochete that causes relapsing fever (RF) and is genetically related to *Borrelia burgdorferi* s.l. It is the only *Borrelia* species in the RF group that can be spread by the Ixodes tick. However, there have been no reports of *B. miyamotoi* in Egypt, and the data on LB in camels is scarce. Therefore, the current study is the first molecular study for the detection of Borrelia spp. and *Borrelia miyamotoi* in camels and associated ticks in Egypt. Furthermore, the identification of tick species that feed on camels demonstrates the value of *cox1*-based molecular approaches for identifying tick species. Here, we provide the first insight into the *Borrelia miyamotoi* and *B. afzelii* found in Egyptian camels and related ticks. Thus, it is essential to comprehend the role of the host in transmission and to constantly monitor the emergence of new diseases in Egypt.

**Abstract:**

Tick-borne diseases (TBDs) are emerging and re-emerging infections that have a worldwide impact on human and animal health. Lyme borreliosis (LB) is a severe zoonotic disease caused by the spirochete *Borrelia burgdorferi* sensu lato (s.l.) transmitted to humans by the bite of infected *Ixodes* ticks. *Borrelia miyamotoi* is a spirochete that causes relapsing fever (RF) and is genetically related to *Borrelia burgdorferi* s.l. However, there have been no reports of *B. miyamotoi* in Egypt, and the data on LB in camels is scarce. Thus, the present study was conducted to screen and genetically identify *Borrelia* spp. and *B. miyamotoi* in Egyptian camels and associated ticks using polymerase chain reaction (PCR). Methods: A total of 133 blood samples and 1596 adult hard ticks were collected from *Camelus dromedaries* at Cairo and Giza slaughterhouses in Egypt. Tick species were identified by examining their morphology and sequencing the cytochrome C oxidase subunit 1 (*cox1*) gene. *Borrelia* spp. was detected using nested PCR on the *IGS* (*16S-23S*) gene, and positive samples were genotyped using *16S* rRNA and *glpQ* spp. genes specific for *Borrelia burgdorferi* and *Borrelia miyamotoi,* respectively. The positive PCR products were sequenced and analyzed by phylogenetic tree. Results: Analysis of the *cox1* gene sequence revealed that the adult ticks belonged to three genera; *Hyalomma* (*H*)*, Amblyomma* (*Am*), *and Rhipicephalus* (*R*)*,* as well as 12 species, including *H. dromedarii, H. marginatum, H. excavatum, H. anatolicum, R. annulatus, R. pulchellus, Am. testudinarium, Am. hebraeum, Am. lipidium, Am. variegatum, Am. cohaerens* and *Am. gemma*. *Borrelia* spp. was found in 8.3% (11/133) of the camel blood samples and 1.3% (21/1596) of the ticks, respectively. Sequencing of the *IGS* (*16S*-23S) gene found that *B. afzelii*, detected from *H. dromedarii* and *H. marginatum,* and *B. crocidurae,* which belongs to the RF group, was detected from one blood sample. *B. burgdorferi* and *B. miyamotoi* were discovered in the blood samples and tick species. Phylogenetic analysis of the *glpQ* gene showed that the *B. miyamotoi* in this study was of the Asian and European types. Conclusions: These results suggest that the camels can be infected by Lyme borrelia and other *Borrelia* bacteria species. This study also provides the first insight into the presence of *Borrelia miyamotoi and B. afzelii* DNA in camels and associated ticks in Egypt.

## 1. Background

The world population of *Camelus dromedaries* (one-hump dromedary camel) is estimated to be over 30 million, with Africa and the Middle East having the largest populations [1]. They are necessary for milk, meat, leather, transportation, and entertainment including tourism in Egypt. As a result, they play a critical role in the socio-economic development of many countries [2,3]. Because these camels are susceptible to several infectious illnesses, eating camel meat or having contact with them represents a significant source of zoonotic disease [1,4]. Tick infestations and TBDs are the most serious threat to camel health, causing global financial losses due to the geographic expansion of their tick vectors [5,6,7]. *Amblyomma, Haemaphysalis, Hyalomma*, and *Rhipicephalus* are the four genera of hard ticks that affect domestic animals in Egypt [8,9,10]. Moreover, tick species that severely infest dromedary camels in Egypt include *H. dromedarii, H. excavatum, H. marginatum, and H. impeltatum* [10,11,12]. Ticks carry a wide range of zoonotic pathogens, including *Borrelia* spp., which cause Lyme borreliosis (LB) and relapsing fever (RF) in humans [13]. Nearly 20 species within the *Borrelia burgdorferi* (sensu lato) complex were included, nine of which are known to cause animal and human LB (i.e., *Borrelia afzelii, Borrelia bavariensis, Borrelia bissettii, B. burgdorferi* (s.s.), *Borrelia garinii, Borrelia kurtenbachii, Borrelia lusitaniae, Borrelia spielmanii and Borrelia valaisiana*) [14]. Lyme borreliosis, which is spread naturally by *Ixodes* spp. ticks causes serious illness in humans, ranging from relatively benign skin lesions to severe cardiac, rheumatic, and neurologic signs [15,16,17]. *Borrelia burgdorferi* s.l. is widespread and is maintained in nature by various arthropod vectors, mammalian, birds, rats, reptiles, and many other wild species [18,19,20].

*Borrelia miyamotoi* is a spirochete that is genetically related to *Borrelia burgdorferi* s.l. It is the only *Borrelia* species in the RF group that can be spread by the *Ixodes* tick [21,22,23]. *Borrelia miyamotoi* infection commonly manifests as a febrile illness accompanied by fatigue, headache, chills, myalgia, arthralgia, and nausea, with potentially fatal complications such as meningoencephalitis [22,24,25,26]. Ticks of all stages are possible vectors of *B. miyamotoi* because it has transstadial and transovarial transmission which is not the case with Lyme borrelia, indicating that the global distribution of *B. miyamotoi*-infected ticks may exceed that of *B. burgdorferi*-infected ticks [27,28]. *B. miyamotoi* has been divided into three groupings based on geographic areas and principal vector species: Asian (or Siberian) (transmitted by *I. persulcatus* and *I. pavlovskyi*), American (transmitted by *I. scapularis* and *I. pacificus*), and European (carried by *I. ricinus*) [29]; and a new fourth clade was recently found in *I. ovatus* ticks in Japan [30,31]. In Africa, RF is most common in the northern part of Africa and is caused by various *Borrelia* spp. such as *B. hispanica, B. duttonii, and B. crocidurae* [32]. However, a new species identified as *B. miyamotoi* has lately sparked renewed attention in this bacterial group [33]. These diseases are challenging to diagnose due to the nonspecific nature of the febrile illness, isolation difficulties, and cross-reactivity between serological techniques [12]. Therefore, it is critical to understand host transmission and to monitor for the emergence of new diseases [34]. *Borrelia miyamotoi* and other relapsing fever group members are distinguished by a glycerophosphodiester phosphodiesterase (*glpQ)* gene [22]. *B. burgdorferi* s.l. lacks the gene for *glpQ*; therefore, this gene is used to detect *B. miyamotoi.* In both *B. burgdorferi* s.l. and *B. miyamotoi,* an intergenic spacer (*IGS*) between the *16S* and *23S* genes is usually used to detect them [35,36].

There have been no reports of *B. miyamotoi* in Egypt, and information on LB in camels is limited. Thus, the present study was conducted to screen and genetically identify *Borrelia* spp. and *B. miyamotoi* in Egyptian camels and associated ticks using molecular methods. Furthermore, to investigate the status of hard tick species that infest Egyptian camels.

## 2. Methods

### 2.1. Sample Collection and Preparation

A total of 133 healthy one-humped camels (*Camelus dromedarius*) aged 3–5 years were investigated for tick infestation. Blood samples and hard ticks were collected from 50 and 83 healthy dromedary camels at Cairo and Giza slaughterhouses in Egypt, respectively. The study was conducted from February 2021 to November 2021. Blood samples (5 mL) were collected in tubes coated with EDTA from the jugular blood vessels of the examined camels, then transferred to the laboratory in an icebox and maintained at −20 °C until DNA extraction. A total of 1596 adult ticks (600 from Cairo and 996 from Giza) were carefully gathered from the camels, then transferred alive to the zoonoses laboratory. Adult ticks were washed twice in distilled water, dried with paper tissues, and the classification of all collected ticks to genus level was by morphological characteristics using a stereomicroscope and taxonomic keys [37].

### 2.2. Extraction of DNA from Blood Samples and Ticks

Morphologically identified ticks (3–5 ticks/genus) were crushed into small pieces in a mortar with liquid nitrogen, then DNA was extracted from the ticks and blood samples (200 µL) using a Thermo Scientific GeneJET Genomic DNA Purification Kit (ThermoFisher, Darmstadt, Germany) according to the manufacturer’s recommended protocol. Isolated material was stored at −20 °C until further molecular analysis.

### 2.3. Molecular Identification of Tick Species

Ticks were identified to species level by amplifying and sequencing an ~820 bp fragment of the cytochrome oxidase c subunit I (*cox1*) gene. Amplification conditioning was performed according to [38]. Table 1 lists the primers used in this study.

### 2.4. Molecular Identification of the Borrelia Species

A nested PCR to detect *Borrelia* spp. in ticks and camel blood samples was carried out using outer and inner primers to amplify the *16S-23S* rRNA intergenic spacer *(16S-23S IGS*) [40], and the amplification was performed according to [36].

Positive samples of *Borrelia* spp. were examined for detection of *B. burgdorferi* using a conventional PCR with a primer set of BbF and BbR for the *16S* rRNA gene, according to [39]. Positive samples were further processed for sequencing.

An additional nested PCR test targeting the *glpQ* gene was performed on the positive samples to confirm the presence of *B. miyamotoi* and to detect genospecies. PCR amplicons were sequenced unidirectionally using primer Q3 to confirm the presence of *B. miyamotoi* in PCR-positive ticks and blood DNA samples [35,36].

The primers and amplification conditions for each reaction are displayed in Table 1. For amplification reactions cosmo Taq DNA Polymerase master mix (Willowfort, UK) was used. Each set of reactions included a positive control and the negative control consisted of nuclease-free water added to the PCR mix instead of the DNA sample. PCR products were visualized on 1.5% agarose gels.

### 2.5. Sequencing and Phylogenetic Analysis

PCR and nested PCR products were purified using a QIAquick purification extraction kit (Qiagen, Hombrechtikon, Switzerland), and sequenced using the BigDye Terminator V3.1 sequencing kit (Applied Biosystems, Waltham, MA, USA). The sequences were assembled into contigs using ChromasPro software (ChromasPro 1.7, Technelysium Pty Ltd., Tewantin, Australia). Next the sequences were aligned with the reference sequences available in the GenBank by BLAST and analyzed using MEGA software X. The phylogenetic trees were constructed using a maximum likelihood (ML) algorithm in MEGA X software using 1000 bootstrap replicates.

## 3. Results

### 3.1. Tick Species

Table 2 shows that 1596 ticks were collected and identified from 133 one-humped camels at Cairo and Giza slaughterhouses. Ticks were collected from healthy camel heads, necks, forelegs, udders, abdomens, back legs, and tails. According to *cox1* gene sequencing, the most common tick species belonged to the *Hyalomma* genus (85.1%). Among tick species, *Hyalomma dromedarii* was the most common species; it accounted for 880 (59.4%), followed by *Hyalomma marginatum* with 297 (20.1%), *Amblyomma hebraeum* with 165 (11.1%), *H. excavatum* with 115 (7.2%), *H. anatolicum* with 66 (4.4%), and *Rhipicephalus annulatus* with 23 (1.6%). A few *Am. testudinarium* 12 (0.8%), *Am. lepidum* 11 (0.7%), *Am. variegatum* 10 (0.7%), *R. pulchellus* 8 (0.5%), *Am. cohaerens* 6 (0.4%), and *Am. gemma* 3 (0.2%) were found.

### 3.2. Prevalence of the Borrelia Species in the Camel Blood Samples and Ticks

The camel blood samples and ticks were screened for the presence of *Borrelia* spp. using nested PCRs with the *IGS* gene and showed that the prevalence of *Borrelia* spp. in adult ticks was 1.3% (21/1596), whereas it was 8.3% (11/133) in the camel blood samples (Table 3). Sequencing of the amplicons revealed the presence of three *Borrelia* spp. in the examined ticks, *B. afzelii* and *B. burgdorferi* (these two genospecies belonged to the *B. burgdorferi s. l.* group), and *B. miyamotoi* (belonging to the RF group). While *B. burgdorferi*, *B. miyamotoi,* and *B. crocidurae* were detected in blood samples (Table 3), *Borrelia afzelii* was not found in blood samples but found in two tick species, while *B. crocidurae* was detected in only one blood sample.

*Borrelia burgdorferi* was confirmed using the *16S*rRNA, revealing that 9.1% (1/11) were positive in blood samples and 14.3% (3/21) were positive in ticks. A second nested PCR on positive blood and tick samples targeting the *glpQ* gene to detect *B. miyamotoi* revealed that *B. miyamotoi* had the highest prevalence in adult ticks [76.2% (16/21)] and camel blood [81.8% (9/11)] (Table 3).

Following the nested PCR and the amplicon sequencing of the *IGS* gene, *Borrelia* spp. were detected in *H. dromedarii* (11/880, 1.3%), *R. annulatus* (2/23, 8.7%)*, Am*. *lepidum* (2/11, 18.1%), *H. marginatum* (1/297, 0.3%), *Am. testudinarium* (1/12, 8.3%), *Am. hebraeum* (1/165, 0.6%), *Am. variegatum* (2/10, 20%), and *Am. cohaerens* (1/6, 16.6%). Additionally, *B. afzelii* was found in *H. dromedarii* and *H. marginatum* (Table 4).

By targeting the *glpQ* gene, *B. miyamotoi* was detected in *H. dromedarii* (8/880, 0.9%), *Am. hebraeum* (1/165, 0.6%), *Am. Lepidum* (2/11, 18.1%), *Am. cohaerens* (1/6, 16.6%), *Am. variegatum* (2/10, 20%), and *R. annulatus* (2/23, 8.7%). While by targeting the *16S* rRNA, *B. burgdorferi* was found in *H. dromedarii* (2/880, 0.2%) and *Am. testudinarium* (1/12, 8.3) (Table 4).

### 3.3. Phylogenetic analysis

Tick species were classified into three genera based on sequence analysis of the positive PCR products of the *cox1* gene: *Hyalomma, Amblyomma, and Rhipicephalus;* this was validated using a sequence identity of 96 to 100% with tick species sequences in GenBank (Figure 1). The accession numbers of *cox1* gene sequences of identified ticks were deposited in GenBank, listed in Table 5.

The phylogenetic tree of *Borrelia* spp. based on the *16S-23S IGS* gene was consistent with the sequence results, identifying four genospecies found in ticks and blood; *B. burgdorferi*, *B. afzelii*, *B. miyamotoi,* and *B. crocidurae*. The *B. miyamotoi* found in *Am. lepidum* and *Am. variegatum* was identical to a German isolate (GenBank: MK945853.1, MK945806.1, MK945787.1, MK458687.1) and clustered with strains from France (MK732472.1) and from Sweden (MK458687.1) which was detected in human cerebrospinal fluid.

The *Borrelia afzelii* sequences in this study clustered in different branches and had significant similarities with German strains (GenBank: MK945805.1). *Borrelia burgdorferi* detected in *Am. testudinarium* clustered with strains detected in *Ixodes pacificus* (MN110090.1, MN110091.1, MN110092.1) in the USA. *B. burgdorferi* was simultaneously isolated from camel blood samples clustered with a strain isolated from humans (KM269456.1) in the USA. *B. crocidurae* found in camel blood clustered with other strains found in a person from France (LT984797.1) and a soft tick from Senegal (KF176328.1) (Figure 2).

The presence of *B. miyamotoi* in ticks and camel blood samples was confirmed using *glpQ* sequence analysis (Figure 3). When compared to each other, most of the studied sequences were 100% identical. They shared a high level of similarity with *B. miyamotoi* sequences found in *Ixodes persulcatus* from Russia (LC538351.1), China (LC557152.1), and Japan (AB900798.1) (Asian type), as well as human blood from Russia (MK955928.1) and China (LC557152.1) (MW319188.1). The *Borrelia miyamotoi* sequence identified in Egyptian camels in *Am. cohaerens* was determined to be identical to *B. miyamotoi* sequences found in *Ixodes ricinus* from Italy (MG451835.1), and Hungary (MF678599.1), as well as in a human CSF fluid from Sweden (MK458689.1) (European type).

Sequence analysis of *B. burgdorferi 16S* rRNA found in camel blood samples and *Am. testudinarium* from Egypt revealed 99% identity with *B. burgdorferi* found in *Ixodes pacificus* from the USA (KY563172.1) and *R. sanguineus* (MH685927.1) and canine blood from Egypt (MH685928.1) (Figure 4).

A list of representative sequences with GenBank accession numbers and their source (tick species or blood) is given in (Table 5).

## 4. Discussion

Ticks and tick-borne diseases (TBDs) have emerged as major public health issues in many countries, including developed ones [41]. Tick infestation in camels has a high economic cost because ticks considerably impact their health and productivity [42]. As the prevalence of tick-borne diseases rises, distinguishing tick species is more critical than ever to improve tick and TBD control [43]. Standard morphological identification might be difficult in blood-engorged, immature, or physically injured specimens. As a result, molecular analysis can aid in discovering new information regarding ticks [44,45]. In this study, an assessment of 133 camels with an ixodid tick infestation revealed 12 species of ticks from 3 genera-infested dromedary camels. There were four species in the *Hyalomma* genus, six in the *Amblyomma* genus, and two in the *Rhipicephalus* genus. The genus *Hyalomma* was the most frequent tick genus in this study (85.1%). *H. dromedarii* is the most common tick species in Egyptian camels, whether locally raised or imported, according to previous studies [10,11,46,47]. *H. dromedarii* infection in camels is dangerous to their health, resulting in significant loss of camels and other animal products in the Middle East and North Africa [48,49], and impacts human health to some extent. Some of the tick species collected in this study (*Am. lepidum, Am. variegatum, H. excavatum, R. pulchellus, and Am. gemma*) are not indigenous to Egypt. This was observed in other studies [10,46,47]. These results might be attributed to camel imports to Egypt’s marketplaces coming from Sudan, Ethiopia, Nigeria, and Somalia [10], suggesting that these camels could be transporting non-endemic tick species to Egypt.

GenBank BLAST using the sequences of the *cox1* genes, validated tick species identification for three genera: *Hyalomma*, *Amblyomma*, and *Rhipicephalus*, and was identical to tick species in the GenBank database. According to the phylogenetic analysis (Figure 1), the *cox1* gene is a valuable and accurate tick species identification marker [50]. Lyme disease is one of the most serious zoonotic diseases, with endemic areas in Central Asia, the United States, and Eastern Europe [51]. This disease has been neglected in the Egyptian camel population.

In the present study, the prevalence rate of *Borrelia* spp. (*B. burgdorferi*, *B. miyamotoi*, and *B. crocidurae)* was detected in camel blood samples using nested PCRs based on the *IGS* (*16S-23S* rRNA) gene (8.3%, 11/133). This result was greater than that found in a study of dromedary camels from Tunisia (1.3%, 3/232) [52] and Bactrian camel blood examined in China (3.6%, 5/138) [20]. In contrast, *Borrelia* spirochete DNA was not found in Iran because the bacteria does not remain in the blood for prolonged periods after infection [53]. In addition, the total prevalence of *Borrelia* spp. in ticks was 1.3% (21/1596). This result was lower than the prevalence of *Borrelia* spp. in *Ixodes* ticks from northern Germany (31.6%, 3150 individual ticks) [54]. While in Ethiopian soft ticks the Prevalence of *Borrelia* spp. was 3.5% (11/312) [55].

Borrelia spp. was detected in *H. dromedarii*, *R. annulatus*, *Am. lepidum*, *H. marginatum*, *Am. testudinarium, Am. hebraeum, Am. variegatum*, and *Am. cohaerens*. This finding matches that of [56,57], who found *B. burgdorferi* in Egyptian Ixodid ticks (*R. annulatus, H. dromedarii, H. excavatum, and R. sanguineus*) and soft ticks (*Ornithodoros savignyi*). Borrelia spp. was detected in Egypt for the first time in *Am. testudinarium*, *Am. cohaerens, Am. hebraeum, Am. variegatum, and Am. lepidum*. This is comparable to a discovery from Korea [58], where they were the first to confirm *B. burgdorferi* in *Am. testudinarium*.

This investigation found *Borrelia afzelii in H. dromedarii and H. marginatum*. The *B. afzelii* sequences found in this study were very similar to isolated strains from Germany, where *B. afzelii* is the most common genospecies in Europe [59]. In Europe, rodents serve as reservoir hosts for *B. afzelii* (e.g., mice and voles) [60]. This infection causes Lyme disease, characterized by acrodermatitis chronica atrophicans and chronic skin disease [61].

While *Borrelia crocidurae* was found in one blood sample, it usually causes a tick-borne relapsing fever (TBRF) transmitted to humans through the bites of soft ticks of the genus *Ornithodoros*. Its impact on public health is just being realized, and it remains an unrecognized and neglected disease [62,63]. These findings suggest that these camels and associated ticks may be infected with Lyme borrelia and other *Borrelia* species, posing a risk to humans.

The presence of *B. burgdorferi* was confirmed using *16S* rRNA gene sequence analysis. Only one sample of *B. burgdorferi* was detected in camel blood and isolated from two *H. dromedarii* and one *Am. testudinarium*. According to the phylogenetic analysis, the Egyptian and United States strains are related. This could be related to bird migration, which aids in the spread of Lyme disease by dispersing *B. burgdorferi*-infected ticks across the country and introduces a new endemic foci [64,65].

TBRF is underdiagnosed in tropical areas due to diagnostic confusion with malaria [66]. *B. miyamotoi* is a spirochete that causes a relapsing fever; spread worldwide by hard *Ixodes* tick species. In 1994, *B. miyamotoi* was isolated from *Ixodes persulcatus* and in 2011, human *B. miyamotoi* illness was first identified in Russia [67]. It is frequently recognized as a human pathogen [28,31]. However, no human *B. miyamotoi* infections have been documented in Egypt. In our study, *B. miyamotoi* was identified in 6.8% of the camel blood samples using the *glpQ* gene.

*B. miyamotoi* has been found in six hard tick species, including *H. dromedarii, Am. hebraeum, Am. lipidium, Am. variegatum, Am. cohaerens* and *R. annulatus*. The prevalence of *B. miyamotoi* in ticks (1%) is comparable to that discovered in Northeast China (1.3% of 774 ticks) with rates of 2.6% in *I. persulcatus*, 0.78% in *Dermacentor nuttalli*, 1.3% in *D. silvarum*, and 0.4% in *Haemaphysalis longicornis*. [28], but the prevalence is lower than that reported from Slovak Republic rodent-attached ticks (3.4%, 31/900) [68]. *B. miyamotoi* DNA was detected in the blood of one camel and in the *H. dromedarii* that infested this camel during our investigation. Sequencing using accession numbers OL347931 and OL439927 corroborated the findings. The results indicate that *Borrelia* was transmitted to the tick during camel bacteremia or that a *Borrelia*-infected tick transmitted the infection to this camel during a blood meal [69]. In the results of our study’s sequences of the *glpQ* gene, our *B. miyamotoi* was genotyped into Asian and European types and was highly similar to sequences of the gene amplified from ticks and human patients. Hence, individuals in Egypt are at risk of developing relapsing fever transmitted by ticks. Unfortunately, the role of camels is still unknown. Similarly, the competence of ticks as vectors for this pathogen needs to be confirmed. More research is needed to obtain more reliable information on the role of camels and associate ticks in pathogen transmission.

## 5. Conclusions

In conclusion, this study sheds light on the tick species that feed on camels in Egypt and demonstrates the value of *cox1*-based molecular approaches for identifying tick species. The camels in Egypt harbor several neglected, emerging, and re-emerging TBDs, many of which are likely new to Egypt, where *Borrelia miyamotoi* and *Borrelia afzelii* were detected for the first time in both camel blood and ticks. Future research needs to comprehend the role of camels in the enzootic cycle of Lyme borrelia.

## Figures and Tables

**Figure 1 vetsci-10-00141-f001:**
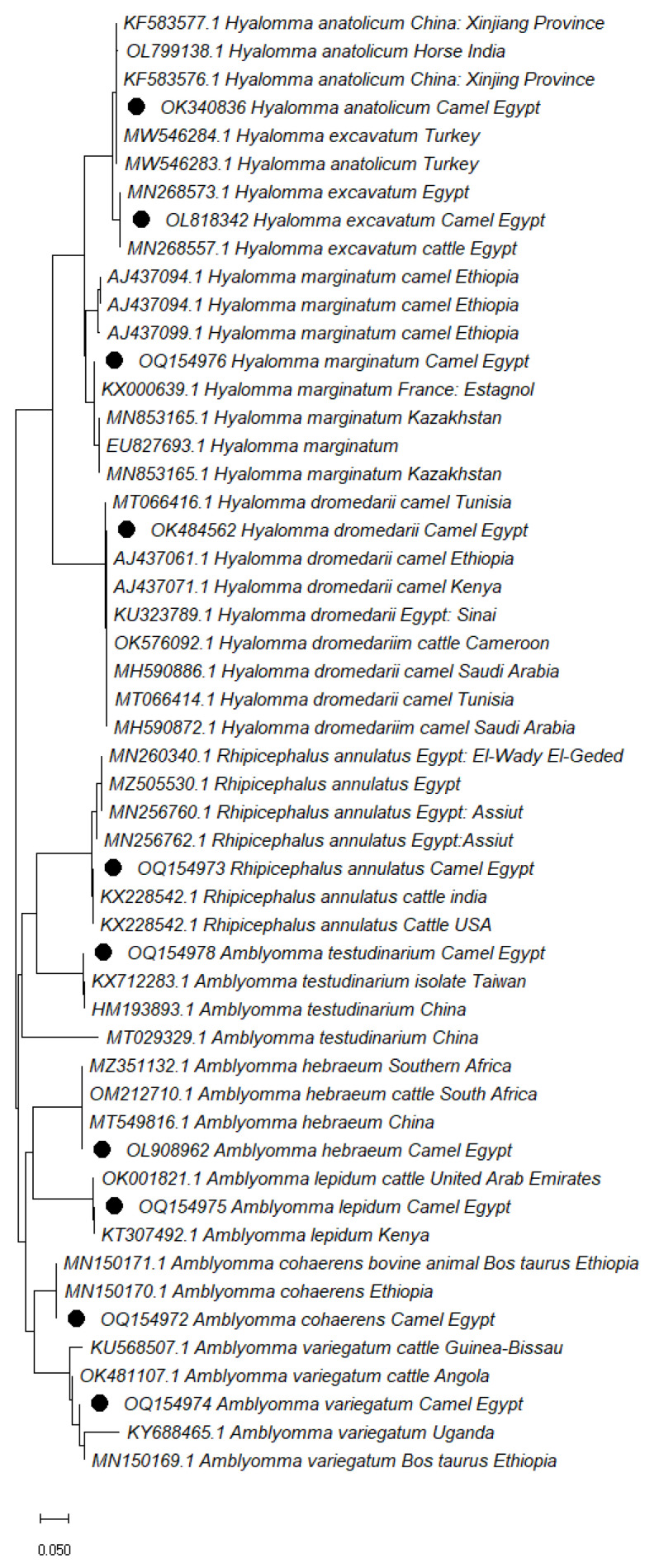
Phylogenetic relationship of tick species collected from camels based on cytochrome c oxidase subunit 1 (*cox1*) gene. The accession numbers with black dots are from this study.

**Figure 2 vetsci-10-00141-f002:**
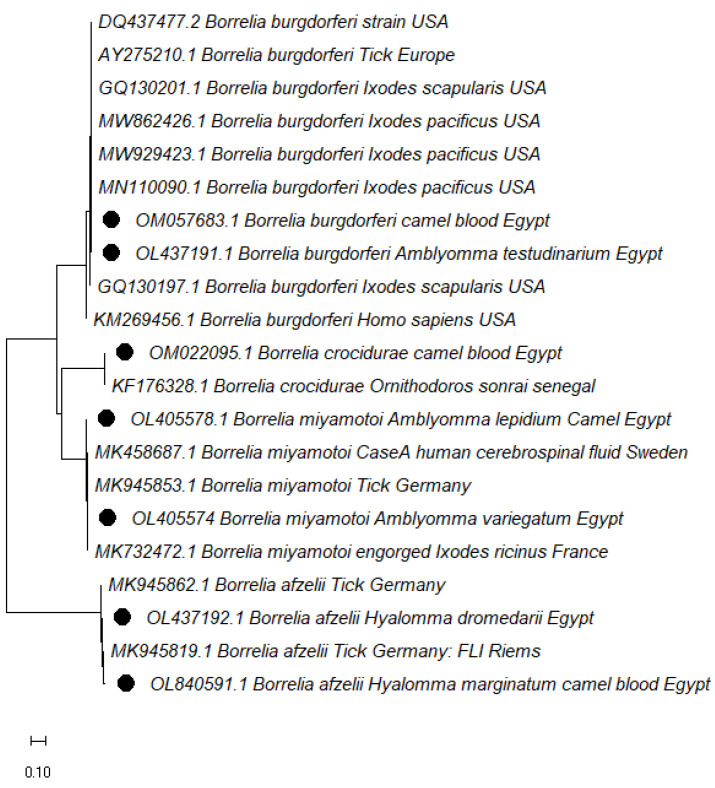
Phylogenetic analysis using Maximum Likelihood method. The evolutionary history of *Borrelia* spp. isolates, based on the *16S-23S IGS*. The accession numbers with black dots are from this study.

**Figure 3 vetsci-10-00141-f003:**
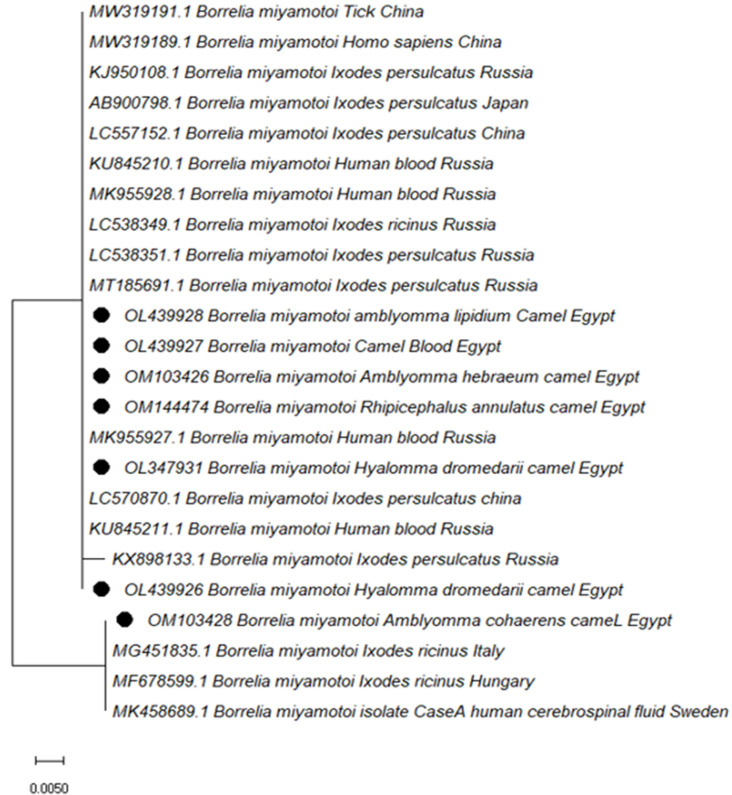
Phylogenetic relationships based on the *glpQ* gene sequences of *B. miyamotoi* the trees were constructed and analyzed using the Maximum Likelihood method. A black dot indicates the new sequences provided by the present study.

**Figure 4 vetsci-10-00141-f004:**
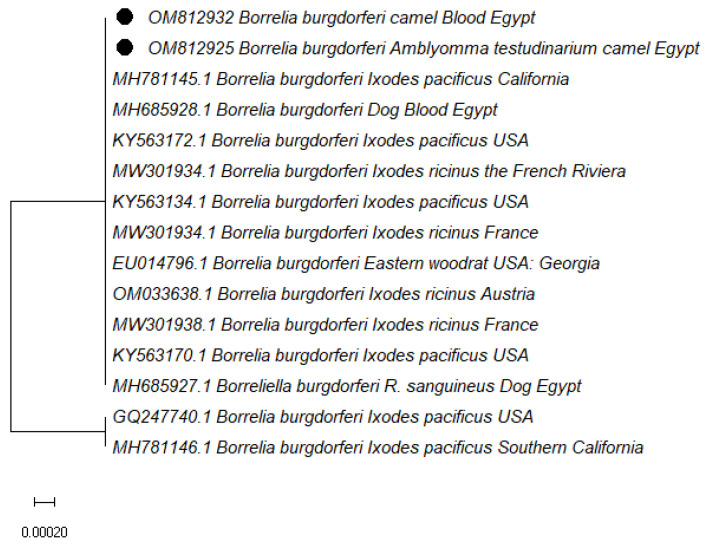
Phylogenetic relationships based on the *16S* rRNA gene sequences of *B. burgdorferi.* The trees were constructed and analyzed using the Maximum Likelihood method. The accession numbers with black dots are from this study.

**Table 1 vetsci-10-00141-t001:** Primer sequences used in this study and PCR amplification conditions.

Pathogen & Tick	Target Gene (bp)	Primer	Sequence (5′-3′)	Thermal Cycles	References
*Borrelia spp.*	*16S*-*23*S rRNA (1007 bp)	Bospp-IGS-F	GTATGTTTAGTGAGGGGGGTG	95 °C, 7 min; 35 cycles (95 °C 45 s; 55.5 °C, 1 min; 72 °C, 45 s), 72 °C, 7 min	[35,36]
		Bospp-IGS-R	GGATCATAGCTCAGGTGGTTAG		
	*16S-23S* rRN*A* (300–600 bp)	Bospp-IGS-Fi	AGGGGGGTGAAGTCGTAACAAG		
		Bospp-IGS-Ri	GTCTGATAAACCTGAGGTCGGA		
*B.burgdorferi*	*16S* rRNA (577 bp)	BbF	GGGATGTAGCAATACATTC	94 °C, 1 min; 35 cycles (95 °C 1 min; 50 °C, 1 min; 72 °C, 1.5 min), 72 °C, 10 min	[39]
		BbR	ATATAGTTTCCAACATAGG		
*B. miyamotoi*	*glpQ* (424 bp)	Q1	CACCATTGATCATAGCTCACAG	95 °C, 5 min; 35 cycles (95 °C 30 s; 50 °C,45 s; 72 °C, 45 s), 72 °C, 7 min	[35,36]
		Q2	CTGTTGGTGCTTCATTCCAGTC		
		Q3	GCTAGTGGGTATCTTCCAGAAC	Annealing: 52 °C	
		Q4	CTTGTTGTTTATGCCAGAAGGGT		
Tick identification	*cox1* (732–820 bp)	CO1-F	GGAACAATATATTTAATTTTTGG	94 °C, 5 min; 30 cycles (94 °C 1 min; 45 °C, 1 min; 72 °C, 1 min), 72 °C, 10 min	[38]
		CO1-R	ATCTATCCCTACTGTAAATATATG		

**Table 2 vetsci-10-00141-t002:** Tick species collected from dromedary camels in Cairo and Giza slaughterhouses of Egypt.

Tick Species	No. (%)
*H. dromedarii*	880 (59.4)
*H. marginatum*	297 (20.1)
*H. excavatum*	115 (7.2)
*H. anatolicum*	66 (4.4)
* **Total/genera** *	**1358 (85.1%)**
*R. pulchellus*	8 (0.5)
*R. annulatus*	23 (1.6)
* **Total/genera** *	**31 (1.9%)**
*Am. hebraeum*	165 (11.1)
*Am. testudinarium*	12 (0.8)
*Am. lepidum*	11 (0.7)
*Am. variegatum*	10 (0.7)
*Am. cohaerens*	6 (0.4)
*Am. gemma*	3 (0.2)
* **Total/genera** *	**207 (13.0%)**
**Total No collected**	**1596**

**Table 3 vetsci-10-00141-t003:** Prevalence of *Borrelia* species in camel blood samples and ticks.

*Borrelia* spp.	Positive Samples	
	Camels Blood	Ticks
*B. burgdorferi*	1 (9.1)	3 (14.3)
*B. afzelii*	0 (0.0)	2 (9.5)
*B. crocidurae*	1 (9.1)	0 (0.0)
*B. miyamotoi*	9 (81.8)	16 (76.2)
Total positive	11 (8.3)	21 (1.3)

**Table 4 vetsci-10-00141-t004:** The prevalence of *Borrelia spp., B. burgdorferi, B. miyamotoi* and *B. afzelii* among the tick Species.

Tick Species	Total No. Ticks Examined	Total *Borrelia* spp. /One Tick spp.	*B. afzelii* (IGS)	*B. burgdorferi* (*16S* rRNA)	*B. miyamotoi* (*glpQ*)
*H. dromedarii*	880	11 (1.3)	1 (0.1)	2 (0.2)	8 (0.9)
*H. marginatum*	297	1 (0.3)	1 (0.3)	-	-
*H. excavatum*	115	-	-	-	-
*H. anatolicum*	66	-	-	-	-
*Am. hebraeum*	165	1 (0.6)	-	-	1 (0.6)
*Am. testudinarium*	12	1 (8.3)	-	1 (8.3)	-
*Am. lepidum*	11	2 (18.1)	-	-	2 (18.1)
*Am. variegatum*	10	2 (20.0)	-	-	2 (20.0)
*Am. cohaerens*	6	1 (16.6)	-	-	1 (16.7)
*Am. gemma*	3	-	-	-	-
*R. annulatus*	23	2 (8.7)	-	-	2 (8.7)
*R. pulchellus*	8	-	-	-	-
Total No.	1596	21 (1.3)	2 (0.1)	3 (0.2)	16 (1.0)

**Table 5 vetsci-10-00141-t005:** Accession numbers of Tick Species and *Borrelia* spp. in this study.

*Borrelia* spp.	Accession Number	Isolation Source	Percent Identity	Target Gene
*B. burgdorferi*	OL437191	*Am. testudinarium*	AF139510 (99.68%)	IGS (*16S*-23S)
*B. afzelii*	OL437192	*H. dromedarii*	OL840591 (99.14%)	
	OL840591	*H. marginatum*	MK945805 (87.90%)	
*B. miyamotoi*	OL405578	*Am. lepidum*	LC540659 (98.93%)	
	OL405574	*Am. variegatum*	CP046389 (97.35%)	
*B. crocidurae*	OM022095	Blood	KF176330 (100.00%)	
*B.burgdorferi*	OM057683		KM269456 (99.74%)	
*B.burgdorferi*	OM812932	Blood	OM033638 (100.00%)	*16S*RNA
	OM812925	Am. testudinarium		
*B. miyamotoi*	OL347931 OL439926	*H. dromedarii*	KJ003841.2 100.00%	*glpQ* gene
	OL439928	*Am. lipidium*		
	OM103428	*Am. cohaerens*		
	OM103426	*Am. hebraeum*		
	OM144474	*R. annulatus*		
	OL439927	*Blood*		
**Tick Species**	**Accession number**	**Isolation source**	**Percent identity**	**Target gene**
*H. dromedarii*	OK484562	Hard ticks	KU323789 (100.00%)	*cox1*
*H. marginatum*	OQ154976		KX000644 (99.87%)	
*H. excavatum*	OL818342		MZ505538 (100.00%)	
*H. anatolicum*	OK340836		KF583576 (100.00%)	
*Am. hebraeum*	OL908962		MT549816 (100.00%)	
*Am. testudinarium*	OQ154978		KX712284 (99.83%)	
*Am. lepidum*	OQ154975		KT307492 (100.00%)	
*Am. variegatum*	OQ154974		MN150169 (100.00%)	
*Am. cohaerens*	OQ154972		MN150171 (100.00%)	
*Rh. annulatus*	OQ154973		KX228542 (100.00%)	

## Data Availability

All the data generated or analyzed in this study are included in this published article.

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
