# Peer review of "The Surveillance of *Borrelia* Species in *Camelus dromedarius* and Associated Ticks: The First Detection of *Borrelia miyamotoi* in Egypt"

_vetsci, 2023, doi:10.3390/vetsci10020141_

Round 1
Reviewer 1 Report
The manuscript by Ashour et al aims to screen ticks collected from camels for the presence of Borrelia spp., the causal agents of Lyme disease and Relapsing fever. For this, the authors collected 1596 ticks and 133 blood samples from camels in slaughterhouses. The ticks were morphologically and molecularly identified. The blood and tick samples were screened for the presence of Borrelia. The results showed that the ticks from 3 different genera were collected from camels and Borrelia spp. were detected in blood and tick samples collected from camels. It was concluded that the camels and associated ticks in Egypt could be infected with different Borrelia bacteria species including Lyme Borrelia, Borrelia miyamotoi and B. afzelii and associated ticks in Egypt. Overall, the topic is well introduced with a well-defined hypothesis. The work is nicely done with a large sample number and had interesting data. However, there is some confusion in tick identification and phylogenetic analysis, which may need closer attention. Please see my detailed comments below.
General comments:
As per Lines 16-17, tick species were identified by examining their morphology and sequencing the cytochrome C oxidase subunit 1 (cox1) gene. But, as described in Lines 107-110, it is not clear how many ticks (All 1596 ticks or a subsample of them) were identified by molecular identification. Were COX1 amplicons of all ticks identified by sequencing or were they identified on gels based on size? Please provide these details in the methods. Table 1: CO1-R primer is missing from the table.
Line 121: Please describe what positive controls were used in each of these PCR reactions.
How many representative sequences were submitted to the NCBI database? Table 5 shows 9 accession numbers targeting COX1 gene, but the phylogenetic trees (Fig 1 A-C) seem to have more than 9 assuming that the species with circles are from this study. Please correct this as needed.
Fig 1: The major issue I have is with Fig 1. For eg: Fig 1A: It is the phylogenetic tree of Amblyomma species. Other than, OL908962 Amblyomma hebraeum, all the other Amblyomma species from this study are clustered separately away from their respective species within the tree which is a little bit unusal. When, I blasted these sequences (NCBI blast), all of them are highly similar to Hyalomma dromedarii and not to Amblyomma sp. which might be a reason they are forming a separate cluster.
For eg: OK440498.1: Amblyomma testudinarium:
The closest match in NCBI is: KU323789.1: Hyalomma dromedarii cytochrome oxidase subunit 1 (CO1) gene, partial cds; 99.86%
This is also true with OK354324 (Hyalomma marginatum) and OL757480 (Rhipicephalus annulatus) (Fig 1B and 1C). I suggest the authors redo the blast and rerun the phylogenetic analysis combining all sequences from this study (with all the 3 genera) as one tree. I believe it will clear things up a lot. Once this is fixed, please make all the necessary changes in the entire manuscript (text and Tables) as appropriate.
Fig 1: Please correct the Figure legend. As per figures: A) Amblyomma and B) Hyalomma
Table 5: Please add the highest similarity percentage in parenthesis next to each Borrelia spp. Please add the accession numbers of the COX1 gene of the isolation source in parenthesis next to the name of the tick species for easy viewing.
Author Response
Thanks for your review.

Reviewer 2 Report
The manuscript (vetsci-2139289) submitted by Ashour et al. consists of a study developed in two cities from Egypt, concerning the detection and identification, by sequencing, of the Borrelia species in the blood and tick samples collected from camelus dromedarius.
I considered this study very relevant and important in the tick-borne diseases and Borrelia areas, however, major alterations and revisions should be done by the authors, so that the manuscript can be accepted in MDPI-Veterinary Sciences.
My first advice is that the phylogenetic analysis should be repeated since in all the presented have the same problem, which is the Borrelia species identified in this study do not branch with the same species from other countries. I considered it normal that a new branch can be formed with the identified species, but the ancestor has to be the same. For example, in figure 1 the tree “A”, the Amblyomma testudinarim identified by the authors must be closest to the remaining Amblyomma testudinarium, even if it is in a new branch, the ancestor has to be the same. This problem is also seen with Amblyomma lepidum and Amblyomma variegatum. This problem can be due to several factors, like for example: the gene zone considered for phylogeny is not the same between the samples from the present study and the samples from other countries, resulting in an impossible comparison; or the size of the amplicon used in this phylogenetic analysis can also influence the obtained results. So, the authors should align their sequences with the ones from the literature and perform the phylogenetic analysis only with the fragments of the gene that align. Also, before any type of alignment, the sequences obtained should be cleaned by deleting the primers that could also influence the phylogeny.
Concerning each section, my suggestions are:
Abstract:
Some italics the ticks and Borrelia species are missing.
Line 20 - Please refer to how the sequences of each PCR were analyzed.
Line 26 – The Borrelia sequences are not isolated, and the authors repeat this term throughout the manuscript. Isolation is normally associated with a culture medium and the obtention of an isolated culture. Please substitute “isolate or isolation” with another term.
Background:
Line 51/52 – Borrelia burgdorferi sensu lato complex, have more Borrelia species than the ones that the authors mentioned, around 20 species belong to the Borrelia burgdorferi (sensu lato) complex, nine of which with recognized pathogenic outcomes to animals and/or humans (i.e. Borrelia afzelii, Borrelia bavariensis, Borrelia bissettii, B. burgdorferi (s.s.), Borrelia garinii, Borrelia kurtenbachii, Borrelia lusitaniae, Borrelia spielmanii and Borrelia valaisiana) in Palaearctic and Nearctic regions.
Line 53 – Lyme borreliosis is not only transmitted by Ixodes spp. Ticks, since several studies have demonstrated the opposite, so I suggest changing the sentence to: “…which is spread naturally by Ixodes spp. ticks…”
Line 58 – Please add: “That can be spread by the Ixodes tick”
Methods:
I considered that a map pointing out the studied cities can improve the manuscript, also to understand if there are close or not. This can also help the authors to compare if the identified Borrelia species can be geographically separated or not, and that information can be added to the results and discussion sections.
Table 1 – The table needs to be improved, namely: there is an extra line between the Q3 and Q4 primers in the thermal cycles section. Also please change the name of this section to PCR amplification conditions. The reverse primer of the COX 1 target gene is missing.
Results
Line 158 – Substitute “and showed” by “showing”.
Table 3 – Substitute “positive animals” by “positive samples” and add “Camels” before blood.
Line 171/172 – the values are switched, according to table 3.
Table 4 – Were there ticks with multiple Borrelia species? Or each tick had only one Borrelia species? If so, please change the table to include that information.
Phylogenetic analysis: Please analyze all the phylogenetic analyses performed for each gene. The branches between the same species must have the same ancestor. Also, the bootstrap values above 75% should be presented in each phylogenetic tree.
The sequence names obtained in this study should be presented in a homologous way, meaning: The authors sometimes referred to the sequences as “camel blood Egypt” and other times as “blood of camel Egypt”.
Also, why do the authors have Borrelia and Borreliella? Can you explain the difference?
Line 219 – Please remove “case of”, and substitute “in Sweden” by “from Sweden”.
Figure 4 – Remove the caps lock.
Discussion
Line 267 – According to what the authors say in this line, and considering the Borrelia DNA found in the blood samples of the analyzed camels, do the authors considered that the camels were recently infected?
Line 270 – Why do the authors compare with a study from Germany? There are no studies in neighbouring countries?
Line 299 – Include a reference to support the sentence “… B. miyamotoi illness was first identified in Russia”.
Line 303/304 – the results concerning China are confused: 1.3% of 774 ticks, and from these 1,8% were I. Ricinus? Please rewrite it to be more perceptible.
Line 308/309 – Regarding the remaining Borrelia species identified in ticks and blood samples, is there also a connection between the host and the tick, like the one observed with B. miyamotoi?
Conclusion
Line 324/325 – Please substitute “… should include a more investigation of the role…” by “Future research with more studies to comprehend the role of camels in the enzootic cycle of Lyme borrelia, should be performed.”
Remove the italic from “Lyme”.
Author Response
Thanks for your review.

Reviewer 3 Report
The work is based on data obtained from an acceptable number of tick specimens and camel blood samples. Manuscript have been written in an accessible and transparent way. I recommend this manuscript to be published in Veterinary Sciences after minor revision. I have some suggestions present bellow.
Abstract
Line 8: Please, delete Background from the sentence.
Line 9: Tick-borne diseases (TBDs) are emerging and re-emerging infections that
have a worldwide impact for a human and animals health.
Line 10: ,, sensu lato and (s.l.)’’-It is correct, don’t use italics. Only Borrelia burgdorferi is written in italics. Please, correct in the whole manuscript.
Line 14: spp. - It is correct, don’t use italics.
Line 17: Borrelia- italics
Line 19: 16S- in italics (16S rRNA ). Please, correct in the whole manuscript.
Line 19: name of species italics :Borrelia burgdorferi and Borrelia miyamotoi
Line 21-22 and whole manuscript. Please, use italics in genera and name of species.
Line 33: Borrelia burgdorferi; Borrelia miyamotoi. Please, use italics
Background
Line 52-53: Borrelia burgdorferi sensu lato (s.l.)- It’s correct
Line 83-85: The sentence,, Thus, the present study was conducted to screen and genetically identify Borrelia spp. and B. miyamotoi in Egyptian camels and associated ticks by polymerase chain reaction (PCR).’’ change to,, Thus, the present study was conducted to screen and genetically identify Borrelia spp. and B. miyamotoi in Egyptian camels and associated ticks using molecular methods.’’
Methods
Line 99-100: Please delete ,,… then stored at −20°C for DNA extraction.’’ from the sentence.
Line 105: ,,…. according to described protocols.’’ Replace to:,, according to the manufacturer’s recommended protocol’’
Line 105: The sentence ,, Genomic DNA was kept at −20°C until for
their molecular analysis.’’ replace to : ,,Isolated material was stored at −20°C until further molecular analysis.’’
Line 111-112: spp. –It’s correct. Don’t use italics (the same in line 115)
Line 114: Please delete ,,… B. burgdorferi and B. miyamotoi’’ I suggest: ,,2.4. Molecular identification of the Borrelia species. ‘’
Line 117-121: Please, the sentence :,, First-round PCR reactions were carried out in a thermocycler (Techne® prime Thermal Cycler, UK), with a total volume of 25 μl consisting of 3 μl of template DNA from each isolate, 12.5 μl of cosmo Taq DNA Polymerase master mix (Willowfort, UK), 1 μl of 10 pmol of each primer, and 7.5 μl nuclease-free water. Reaction conditions were the same in the first and second rounds, according to [35].’’ Replace to: ,, A 3 μl DNA template was used for the primary reactions and 1 μl of amplification product for the nested amplifications. For both reactions of cosmo Taq DNA Polymerase master mix (Willowfort, UK), was used and the amplification was performed according to [35].’’
Line 121-122: The sentence ,, Each set of reactions included a positive and negative control (molecular grade water)’’ replace to:,,
,,Each set of reactions included a positive control of Borrelia spp. ( add GenBank Acc.No.(If it exist) isolated from…(source) of …(host) , while the negative control consisted of nuclease-free water added to the PCR mix instead of the DNA sample.’’.
Line 122-123: I propose the sentence:,, Amplification products were visualized on
a 1.5% agarose gel, and positive samples were further processed for sequencing.’’ Replace to :,, Nested PCR products were visualized on 1,5% agarose gels stained with (name of solution) (Company, city, country).’
Line 133-135: Change or iprove this fragment of manuscript: ,,The positive samples were purified using a QIAquick purification extraction kit (Qiagen, Hombrechtikon, Switzerland). The purified PCR products were then sequenced using the BigDye Terminator V3.1 sequencing kit (Applied Biosystems, Waltham, MA).’’
I propose:,, PCR and nested PCR products were purified using a QIAquick purification extraction kit (Qiagen, Hombrechtikon, Switzerland), and sequenced using the BigDye Terminator V3.1 sequencing kit (Applied Biosystems, Waltham, MA).’’
Line 135-142: Please, change this fragment of manuscript:
,,The sequences obtained were assembled and edited by ChromasPro software (ChromasPro 1.7, Technelysium Pty Ltd., Tewantin, Australia). The corrected sequences were compared
with the reference sequences available in the GenBank by BLASTN (https://blast.ncbi.nlm.nih.gov/Blast.cgi). The obtained sequences were recorded in GenBank. The obtained sequences and sequences available in GenBank were aligned using
CLUSTAL W in MEGA software X. The phylogenetic trees were inferred using the maxi-
mum likelihood method with 1000 bootstrap replicates in MEGA X software.’’
I suggest:,, The sequences were assembled into contigs using ChromasPro software (ChromasPro 1.7, Technelysium Pty Ltd., Tewantin, Australia). Next the sequences were aligned with the reference sequences available in the GenBank by BLAST and analyzed using in MEGA software X. The phylogenetic trees were constructed using a maximum likelihood (ML) algorithm in MEGA X software using 1000 bootstrap replicates’’
Results
I suggest mention and adding morphometric results to this section. Morphometric study are mentioned in the methods section but no information provided in the results section. I think that morphometric identification of ticks is generally sufficient…..
Line 144: change to: ,, 3.1 Tick species’’
Line 149: Please, use the full name of the species if you mention it for the first time in the manuscript. Correct on: Hyalomma dromedarii. It the same in Line 150-153.
Line 156: I suggest the section 3.2 and 3.3 (line 173) put under the one heading: ,, 3.3.
,, Prevalence of the Borrelia species in the blood samples and ticks’’.
Line 161: The sentence ,, Table 3. Detection of Borrelia species in camel blood samples and ticks by PCR assay.’’ change to :,, Table 3. Prevalence Borrelia species in camel blood samples and ticks ‘’
In table 3 B. Afzelii replace ,, B. afzelii’’
Line 179: B. Spp replace ,, Borrelia spp.’’
In Table 2 and Table 3 insert a space in the name of tick species. ,, H.dromedarii ‘’ change to ,, H. dromedarii’’
Author Response
Thanks for your review.

Round 2
Reviewer 1 Report
The authors reran the phylogenetic analysis and satisfactorily revised the manuscript. There are a few more minor corrections listed below. The line numbers stated below are from the marked version of the revised manuscript.
Table 1: CO1-R primer is still missing from Table 1.
Line 125- 134: This paragraph (2.4: Molecular identification of Borrelia) needs to be rephrased and rearranged. I suggest moving the reaction mix and primers before the template DNA sentence.
Line 128: Please delete “of”. For both reactions, cosmo Taq DNA.
Line 129: Not sure why primer details are removed from this section. Please include it. (eg: 1 µl of respective forward and reverse primers at 10 pmol concentration…)
Table and figure titles need to be elaborated with more information.
Eg: Table 2: Tick species collected from dromedary camels in Cairo and Giza of Egypt in 2021. (or something similar).
Figure 1 title: Phylogenetic relationship of tick species collected from camels based on cytochrome c oxidase subunit 1 (cox1) gene. The accession numbers with black dots are from this study.
This is just an example. This can be elaborated further and please fix the titles for all the figures as appropriate.
Line 252 (Figure 3 title): The dots are black
Figures 1, 2 3, and 4: Please italicize the scientific names in the phylogenetic trees. Scientific names also need to be italicized at several places in the main text (Eg: Line 317 in the marked version)
Author Response
The authors reran the phylogenetic analysis and satisfactorily revised the manuscript. There are a few more minor corrections listed below. The line numbers stated below are from the marked version of the revised manuscript.
Thanks for your comment.
Table 1: CO1-R primer is still missing from Table 1.
We add it
Line 125- 134: This paragraph (2.4: Molecular identification of Borrelia) needs to be rephrased and rearranged. I suggest moving the reaction mix and primers before the template DNA sentence.
Done
Line 128: Please delete “of”. For both reactions, cosmo Taq DNA.
Done
Line 129: Not sure why primer details are removed from this section. Please include it. (eg: 1 µl of respective forward and reverse primers at 10 pmol concentration…)
Done
Table and figure titles need to be elaborated with more information.
Eg: Table 2: Tick species collected from dromedary camels in Cairo and Giza of Egypt in 2021. (or something similar).
Done
Figure 1 title: Phylogenetic relationship of tick species collected from camels based on cytochrome c oxidase subunit 1 (cox1) gene. The accession numbers with black dots are from this study.
Done
This is just an example. This can be elaborated further and please fix the titles for all the figures as appropriate.
Line 252 (Figure 3 title): The dots are black
Done
Figures 1, 2 3, and 4: Please italicize the scientific names in the phylogenetic trees. Scientific names also need to be italicized at several places in the main text (Eg: Line 317 in the marked version)
Done
Reviewer 2 Report
The manuscript have improved and I considered it ready for publication.
Author Response
Thanks for your comment.